# Mindfulness, Loving-Kindness, and Compassion-Based Meditation Interventions and Adult Attachment Orientations: A Systematic Map

**DOI:** 10.3390/bs15020119

**Published:** 2025-01-24

**Authors:** Taranah Gazder, Erica Ruby Drummond, Mine Gelegen, Sarah C. E. Stanton

**Affiliations:** Department of Psychology, University of Edinburgh, Edinburgh EH8 9JZ, UK; ericarubyd@gmail.com (E.R.D.);

**Keywords:** attachment, meditation, mindfulness, loving-kindness, compassion, intervention science, systematic map

## Abstract

Meditation interventions have important benefits, including potentially helping those with higher attachment anxiety and avoidance enjoy better personal and relational growth. This preregistered and reproducible systematic map sought to identify the extent and scope of experimental evidence investigating the role of mindfulness, loving-kindness, and compassion-based multi-session meditation interventions in (a) moderating the effects of attachment orientations on outcomes over time and/or (b) changing attachment orientations over time. We conducted a systematic map, as the literature on meditation interventions and attachment is nascent. We searched 5 databases, screening 725 studies. We extracted data from four journal articles and one dissertation (five studies in total) which met our inclusion criteria. Four studies examined the effects of meditation interventions on buffering attachment insecurity and one examined attachment security enhancement. All five studies included a mindfulness meditation intervention, and one included a loving-kindness meditation intervention. All studies were conducted in Western cultures. Studies primarily found evidence for interventions benefitting those with higher attachment anxiety, although some evidence emerged for higher attachment avoidance. Our systematic map highlights a critical need for further application of meditation interventions in an attachment and personal growth context, given the promising nature of early work in this area.

## 1. Introduction

Meditation interventions have important personal and relational benefits, including improving psychological well-being ([33]), promoting social connection ([28]), enhancing relationship satisfaction and closeness ([9]), and promoting personal growth ([19]; [35]). Meditation interventions are relatively straightforward and inexpensive to administer and do not require personalising, thereby offering cost-, time-, and resource-effective methods to promote positive outcomes for people and communities ([56]). One important area in which meditation interventions may show promise is in (a) buffering against the typically negative personal and relational consequences that stem from attachment anxiety and avoidance (i.e., buffering attachment insecurity, see [15]; [21]), as well as (b) reducing attachment anxiety and avoidance over time (i.e., enhancing attachment security, see [22]; [47]). The purpose of the present research was to systematically map the current experimental literature that has studied meditation interventions and attachment orientations over time. Systematic maps are often used when a field is in its earliest stages to identify knowledge clusters and gaps ([12]) or investigations of combinations of variables ([25]). Unlike systematic reviews or meta-analyses, systematic maps do not aim to assess the effectiveness of interventions as there is typically insufficient evidence to be able to do so ([25]); rather, they aim to catalogue the breadth and scope of evidence in a given domain. We aimed to create a reproducible systematic map which can be updated and extended over time as knowledge on this subject grows.

### 1.1. Attachment Theory

Attachment orientations refer broadly to the way in which we relate to close others and shape relationship, career, and health and well-being outcomes throughout life ([14]; [37]). Attachment orientations develop in infancy based on how our primary caregivers respond to our needs ([5]). The attachment system is an inborn system which motivates us to seek proximity to and comfort from close others in times of distress ([5]; [37]). However, the attachment system does not always function optimally, and the way in which it operates is shaped by the way in which our primary caregivers respond to our needs in infancy and childhood, leading to individual differences in attachment orientations ([5]).

In adulthood, attachment orientations fall along two theoretically distinct dimensions: attachment anxiety and attachment avoidance ([6]; [20]). High levels of attachment anxiety develop when caregivers are inconsistent in responding to needs ([5]). This creates uncertainty about whether their needs will be met and leads to highly anxious individuals trying hard, sometimes to the detriment of themselves and their close others, to gain or regain closeness. Attachment anxiety is characterised by worries about abandonment and rejection, rumination about relationships, and hyperactivating behavioural strategies such as perceiving more conflict in relationships and escalating the severity of relationship conflict ([8]; [37]). Meanwhile, high levels of attachment avoidance develop when caregivers consistently ignore, dismiss, or do not respond to needs, which in turn leads to the belief that it is safer to depend only on oneself ([5]). Attachment avoidance is characterised by discomfort with closeness and intimacy, desire for emotional distance from relationship partners, and deactivating behavioural strategies such as withdrawing from conflict and support experiences in relationships ([37]; [40]). Individuals who grow up with caregivers who generally respond to their needs and show care towards them develop attachment security ([5]). Attachment security is represented by low scores on attachment anxiety and/or avoidance; these individuals tend to be comfortable with both independence and interdependence, and they enjoy the most positive outcomes and growth ([37]).

Although attachment orientations are shaped in infancy and childhood, they can shift over time in response to several factors, including therapy and positive experiences within close relationships ([3]; [10]). People scoring lower on attachment anxiety and/or avoidance tend to have the most positive personal and relational outcomes, making the identification and implementation of experiences and interventions that enhance attachment security an important goal for researchers. There are two primary ways in which experiences and interventions targeting attachment orientations can operate. First, these experiences and interventions can nullify or lessen the typically negative links between attachment anxiety and avoidance and outcomes (e.g., relationship satisfaction, personal growth, health perceptions), which is termed attachment insecurity buffering. Alternately (or in addition), these experiences and interventions can “get to the root” of the struggles inherent in attachment insecurity and ultimately reduce levels of attachment anxiety and avoidance themselves, which is termed attachment security enhancement ([3]).

A growing body of the literature has examined various processes through which attachment insecurity (i.e., attachment anxiety and avoidance) can be buffered and/or attachment security can be enhanced. Specifically, the Attachment Security Enhancement Model (ASEM; [3]) theorises that partners can engage in tailored strategies which can buffer attachment anxiety and avoidance, while different tailored strategies can reduce each form of attachment insecurity. With regards to buffering attachment insecurity, the ASEM suggests that attachment anxiety can be buffered via safe strategies, which involve conveying a strong, intimate emotional bond and providing reassurance while at the same time deescalating negative emotions ([3]; [39]). Meanwhile, the ASEM theorises that attachment avoidance can be buffered via soft strategies, behaviours which are sensitive to an avoidant person’s discomfort with emotional interactions and intimacy. These strategies may include communicating why certain needs and requests are reasonable and respecting the avoidant individual’s need for autonomy and space within the context of positive relationship experiences ([3]; [40]).

Turning to enhancing attachment security, the ASEM ([3]) postulates that attachment anxiety should decline in situations which foster greater personal confidence and more secure models of the self. For example, trait general mindfulness predicts decreases in attachment anxiety over time ([21]), possibly by redistributing attention away from a hyperfocus on close others and perhaps towards a more evenly spread attention on one’s own goals and interests. Conversely, the ASEM suggests that attachment avoidance should decline in situations which involve positive dependence and more secure working models of close others. For instance, daily positive relationship events and intimacy promotion interventions predict declines in attachment avoidance ([48]). However, research on cost-efficient, effective meditation interventions to buffer attachment insecurity and enhance attachment security is rare.

### 1.2. Meditation Interventions and Adult Attachment Orientations

Meditation interventions are likely to be helpful for individuals with higher attachment anxiety and/or avoidance. Meditation interventions come in many varieties ([44]). In the current research, we focused specifically on mindfulness, loving-kindness, and compassion-based meditation interventions. Mindfulness meditation trains individuals to focus attention on present-focused thoughts, feelings, and experiences in an accepting manner ([13]; [17]). It originated from Buddhist traditions but today is often used in secular contexts ([55]). Mindfulness meditation entails sitting quietly and is characterised by aiming to observe one’s experiences in the present moment in a non-judgmental way ([17]). The exact object of these observations (e.g., breath, bodily sensations, thoughts, feelings, sounds) can differ across different types of mindfulness meditation ([17]). Mindfulness meditation has been associated with several positive outcomes, including relationship satisfaction, closeness, and acceptance of others ([9]).

Loving-kindness meditation involves individuals directing warm, kind-hearted thoughts towards specific targets, increasing the capacity for empathy and compassion ([45]). The core psychological operation of generating kind intentions towards one’s targets is consistent across all loving-kindness meditations, but the exact procedures can differ. Generally, practitioners of loving-kindness meditation silently repeat certain phrases (e.g., “May you be happy”, “May you be healthy”) towards targets, which range from oneself to a close other, a neutral other, a disliked other, and to the world at large ([55]). In some traditions, practitioners also use imagery, visualising the target or imagining light flowing from one’s heart towards others ([52]). Like mindfulness meditation, loving-kindness meditation originated from Buddhist traditions but is now practiced in secular contexts across the world ([55]). Loving-kindness meditation has been linked to increases in positive emotions, self-compassion, and life satisfaction ([23]; [34]; [55]).

Finally, compassion-based meditations focus awareness on the alleviation of the suffering of all beings. Compassion-based meditations are similar to loving-kindness meditations, and the two are often combined both in traditional Buddhist practices and in psychological studies today ([27]). Similar to loving-kindness meditation, compassion-based meditation employs the imagining or actual experience of the desired state (e.g., warm-heartedness, compassion) as the object of attention ([27]). Compassion-based meditations are focused on cultivating a desire to understand and assuage others’ suffering. Compassion-based meditation has been found to be effective in promoting happiness and decreasing worry and emotional suppression ([29], [30]), as well as enhancing empathic accuracy ([36]).

There is substantial cross-sectional and some longitudinal evidence demonstrating links between the constructs that meditation interventions frequently target (e.g., mindfulness, compassion) and attachment anxiety and avoidance. For example, one meta-analysis found moderate negative associations between mindfulness and both attachment anxiety and avoidance ([50]). Other research has found moderate negative associations between compassion and both attachment anxiety and avoidance ([4]; [38]). Non-experimental research has shown that mindfulness can buffer the typically negative links between attachment anxiety and avoidance and personal and relationship outcomes such as sexual motivations and relationship behaviours ([15]; [21]). There is also evidence that general and relationship-specific mindfulness can reduce attachment anxiety and avoidance over time ([22]) This is also theoretically consistent; following the logic of the ASEM ([3]), mindful attention may reduce attachment anxiety over time by decreasing preoccupation and worry about close relationships, drawing focus to the present moment ([22]). Meanwhile, mindful attention may reduce attachment avoidance over time by helping avoidant individuals be more present in intimate moments with close others, helping them feel more empathy and connection with close others ([22]). However, experimental studies of meditation interventions buffering attachment insecurity and enhancing attachment security are in their infancy. Although there has been some work examining the effects of priming state mindfulness and examining effects on state attachment security (e.g., [51]), there is very limited work examining the longitudinal impact of formal meditation interventions, which are more likely to have lasting insecurity-buffering or security-enhancing effects. In the current research, we chose to focus only on interventions which lasted at least two sessions because a single session in the meditation literature is generally considered an “induction”, or priming, rather than an intervention with effects over time (cf. [7]). In summary, given the potential for meditation interventions to impact attachment orientations and the outcomes resulting from attachment orientations, gaining a sense of the current literature is vital for researchers working in these areas.

### 1.3. Present Research Overview

Because the literature on meditation interventions and attachment orientations is nascent, we chose to conduct a systematic map. Systematic maps are a method of evidence synthesis designed to assess the nature and state of a literature base and are used to locate and catalogue the breadth of evidence on a particular topic with the end goal of creating a searchable database of studies ([24]; [25]). Systematic maps are often used when a field is in its early stages to identify knowledge clusters and gaps ([12]) or investigations of combinations of variables ([25]). Unlike systematic reviews or meta-analyses, systematic maps do not aim to answer questions about the effectiveness of interventions, the merit of a hypothesis, or effect sizes, as there is typically insufficient evidence to be able to do so ([25]).

Thus, the purpose of the present research was to systematically identify and map research examining the links between meditation-based interventions and adult attachment orientations. We focused our search on peer-reviewed articles, theses, and dissertations. Our objective was to gauge the extent and scope of experimental evidence indicating that meditation interventions (a) moderate the effects of attachment orientations on personal and relationship outcomes over time and/or (b) change attachment orientations over time.

## 2. Materials and Methods

### 2.1. Systematic Map Protocol

Our preregistered protocol (https://osf.io/qkd65/?view_only=f454bb0f70c847d19523b8705c019e17, accessed on 18 March 2024) and analyses adhered to the Preferred Reporting Items for Systematic Reviews and Meta-Analyses (PRISMA; [41]). Some alterations were made to make these guidelines suitable for a systematic map rather than a systematic review. Specifically, Risk of Bias, Data Synthesis, Meta-Biases, and Confidence in Cumulative Evidence were not assessed, given the small number of studies assessed in this map as well as the fact that systematic maps aim to locate and catalogue the breadth of evidence on a particular topic rather than draw conclusions regarding findings or evaluate the merits of methodology used ([24]; [25]).

### 2.2. Search Term Identification and Selection

We generated a draft search strategy and gradually refined this using an iterative process by performing searches in PsycInfo (Ovid interface), MEDLINE (PubMed), SCOPUS, Web of Science, and ProQuest Dissertations & Theses Global, with database-specific Boolean operators applied. The details of each final search performed can be found in Table 1. The full search strings and search instructions for each database can be found in our online Appendix A at https://osf.io/d29w8?view_only=f454bb0f70c847d19523b8705c019e17, accessed on 27 February 2024. No date restrictions were applied to the searches. Searches were completed between the 19 and 27 February 2024.

### 2.3. Databases

Searches for peer-reviewed papers were carried out on PsycInfo (Ovid interface), MEDLINE (PubMed), SCOPUS, and Web of Science. Searches for theses and dissertations were carried out on PsycInfo, Medline, and ProQuest Dissertations & Theses Global.

### 2.4. Inclusion and Exclusion Criteria

Our detailed criteria for including or excluding texts appear in Table 2.

### 2.5. Selection Process

Publications returned from the electronic searches were imported into Zotero. These records were then uploaded into Covidence, Cochrane’s online systematic review tool which facilitates collaborative screening. Covidence automatically removes duplicates which it can identify. Other duplicates were removed manually. Two authors (TG and ERD) screened the titles and abstracts yielded by our searches against our inclusion and exclusion criteria. Studies were coded as (a) yes (include), (b) no (do not include), or (c) maybe (maybe include, pending further information). Papers were coded as “maybe” if there was insufficient information available to make a final decision regarding inclusion. In “maybe” cases, the papers were retained and reviewed at full-text screening. The coders undertook calibration exercises for the first 20 screened articles to ensure consistency. Cohen’s Kappa for title and abstract screening was 0.60, indicating moderate agreement. Covidence highlights discrepancies in a section called Resolve Conflicts. The two coders met to resolve all disagreements by discussion. We then obtained full-text versions for all titles which either appeared to meet the inclusion criteria or where there was ambiguity regarding whether they met the inclusion criteria. The same two coders then screened the full-text versions and made final decisions about whether these met the inclusion criteria as well as providing reasoning in cases of exclusion. Like title and abstract screening, disagreements were resolved by discussion of the Resolve Conflicts section. Cohen’s Kappa for full-text screening was 0.58, indicating moderate agreement. Disagreement was resolved through discussion. The coders were not blind to journal titles, study authors, or institutions.

### 2.6. Data Collection Process and Extraction Framework

Using standardised forms on Covidence, two authors (TG and MG) extracted data from each eligible study. The coders went through calibration exercises prior to data extraction to ensure consistency. Information extracted from each study was discussed by the two coders, and agreement was reached on all information extracted. The information extracted from all relevant studies appears in Table 3.

## 3. Results

In total, we obtained 1340 text results from our database searches, 725 of which were original and continued to title and abstract screening. After the screening, 20 texts continued to full-text screening to determine final eligibility. Out of these, five texts were determined to meet all inclusion criteria. For a complete breakdown of the screening process, see Figure 1. We extracted data from these five texts, four of which were peer-reviewed journal articles and one of which was a PhD dissertation. Four texts reported a single study. The dissertation ([43]) reported two studies, but data were extracted only from the second study, as the first one did not meet our inclusion criteria. All data extracted can be found in Table 4.

## 4. Discussion

The current paper aimed to create a reproducible systematic map assessing the state of the literature on the role of meditation interventions in buffering attachment insecurity and enhancing attachment security. With our search strategy and extraction framework openly available, we hope these findings can be reproduced and extended over time as the literature linking meditation interventions with attachment orientations grows.

### 4.1. Key Findings

Five studies met our criteria, four of which examined attachment insecurity buffering ([11]; [16]; [43]; [54]), and one of which examined attachment security enhancement ([32]). Two studies found evidence only for attachment anxiety buffering on positive sexuality ([16]) and trait mindfulness ([43]). Meanwhile, a single study ([54]) found evidence for both attachment anxiety (on positive and negative emotions in the mindfulness meditation condition) and attachment avoidance buffering (on negative emotions in both the mindfulness meditation and the loving-kindness meditation conditions). One study ([11]) examined attachment insecurity as a composite measure of attachment anxiety and attachment avoidance and found marginal insecurity buffering effect of Mindfulness-Based Stress Reduction (MBSR) on perceived stress. The one study assessing attachment security enhancement ([32]) found reductions in attachment anxiety, but not attachment avoidance, following a variation of MBSR.

All five texts employed some form of mindfulness meditation intervention, while one ([54]) also used a loving-kindness meditation intervention. However, it is important to note that there are meaningful differences between “pure” mindfulness meditation interventions and MBSR ([46]), including their active ingredients ([17]). Mindfulness meditation interventions generally focus only on elements of nonjudgmental attentional awareness, while interventions such as MBSR tend to include a conglomerate of elements (e.g., mindfulness, loving-kindness, psychoeducation, and gratitude). Interventions which include only mindfulness meditation appear to be more closely linked to increases in variables such as trait mindfulness and decreases in anxiety, while MBSR appears to have a far larger effect on reducing negative emotions and increasing well-being ([17]). Given that two out of five studies in the current map used variations of MBSR ([11]; [32]) while the other three employed “pure” mindfulness meditation interventions ([16]; [43]; [54]), these studies are likely not directly comparable. It is more difficult to conclude that mindfulness is the active ingredient in the case of MBSR; MBSR does not exclusively work through mindfulness meditation interventions but also through psychoeducation as well as integration of other meditation interventions such as loving-kindness (albeit in much smaller amounts than mindfulness meditation) ([17]). Therefore, in MBSR-based studies, it is difficult to parse out what the active ingredients are. This points to the importance of dismantling future MBSR interventions to isolate what the active ingredients are (cf. [18]). Isolating different active ingredients (e.g., teaching mindfulness and loving-kindness meditation separately) in separate interventions would help clarify which aspects drive attachment insecurity buffering and attachment security enhancement, bearing in mind that different processes are often responsible for the two and that there are different processes for reducing attachment anxiety and avoidance as well ([3]).

While our systematic map discovered that mindfulness and loving-kindness meditations are potentially viable interventions which may buffer attachment insecurity and/or enhance attachment security, we found no work assessing the impact of compassion-based meditations. It may be that compassion-based interventions play a different role than others (e.g., compassion-based meditation interventions help individuals to form more positive working models of close others). This creates a significant research gap which would benefit from being filled.

The interventions employed in the studies in this systematic map were rigorous. In our inclusion criteria, we outlined that studies would need to include at least two intervention sessions to be included. However, all of the studies included more than this, with a minimum of six sessions used. Some interventions were conducted in person (e.g., [54]), while others were conducted using audio recordings (e.g., [16]). Future research would benefit from examining whether the method of delivery influences the efficacy of the interventions. All studies included also employed a reasonable sample size, highlighting the robustness of results.

In the current studies, there was little long-term follow-up. In four out of five studies included in this map, the only measurement of outcome variables was immediately following intervention. The only exception ([54]) involved participants providing daily reports for three weeks following the interventions. Following participants over longer periods of time is vital when it comes to buffering attachment insecurity as well as enhancing attachment security since these processes often take place over months and years ([2]; [42]). This is, therefore, a crucial direction for future research investigating links between meditation interventions and attachment orientations.

There was cultural bias in the current set of studies; all but [16] ([16]) were conducted in North America, meaning that conclusions need to be drawn carefully (see [26]). This highlights the importance of conducting meditation intervention research in the domain of attachment with more racially and culturally diverse samples, especially considering cultural differences in mindfulness, compassion, and attachment ([1]; [31]; [49]). Furthermore, all the studies in this systematic map had samples in which most participants were female. Female participants tend to practice meditation more and sometimes report differences in their reasons for meditating compared to male participants ([53]). Thus, incorporating more gender-balanced samples (including individuals who identify as nonbinary) and examining possible gender differences will enhance understanding of those whose attachment orientations will benefit from meditation interventions the most.

Moreover, the interventions included in this group of studies varied substantially in terms of design, duration of intervention sessions (between 8 min and 1 h), and duration of interventions (2 to 8 weeks), making comparisons between studies more challenging. Using more consistent structures and durations (e.g., standardised MBSR protocols) to examine these effects would make it easier to draw firmer conclusions in future research. The frequency of intervention varied between daily and weekly. Teasing out differences between these when a larger evidence base has been built up would be a helpful future direction.

### 4.2. Strengths and Limitations of the Systematic Map

We aimed to conduct our systematic map in the most rigorous manner possible, conducting systematic searches on multiple databases and continually refining our search strategy. We also chose to include theses and dissertations in our search to gain a better sense of the upcoming literature in the field which may not yet be published in peer-reviewed journals. In addition, we used broad search terms to gain a robust understanding of the field, including a wide variety of meditation interventions. We are confident that our search strings are reproducible and can be used in the future to gain an up-to-date understanding of the state of the literature.

However, our process is not entirely bias-free. We included only articles published in English in our systematic map. Although a great deal of the international scientific literature is published in English, our inclusion and exclusion criteria may have created a gap where relevant studies written in other languages were not included. Moreover, while we aimed to be as broad as possible in our selection of search terms and employed an iterative process in conducting the searches, it is possible that, as a team of social psychologists, we may have missed some relevant terms in adjacent fields.

## 5. Conclusions

This systematic map highlights the promising literature in its infancy and a critical need for further research into the application of meditation interventions in the context of adult attachment, given the potential to promote well-being and personal growth ([35]; [37]). The results of this systematic map show that there is exciting potential for meditation interventions to play an important role in the buffering of attachment insecurity and the enhancement of attachment security. We recommend that this domain of research can be advanced through conducting studies over longer periods of time, isolating active ingredients (e.g., mindfulness separated from psychoeducation), examining if compassion-based meditation interventions are also viable for buffering or changing attachment, and conducting research across more diverse demographics and in a wider range of cultures.

## Figures and Tables

**Figure 1 behavsci-15-00119-f001:**
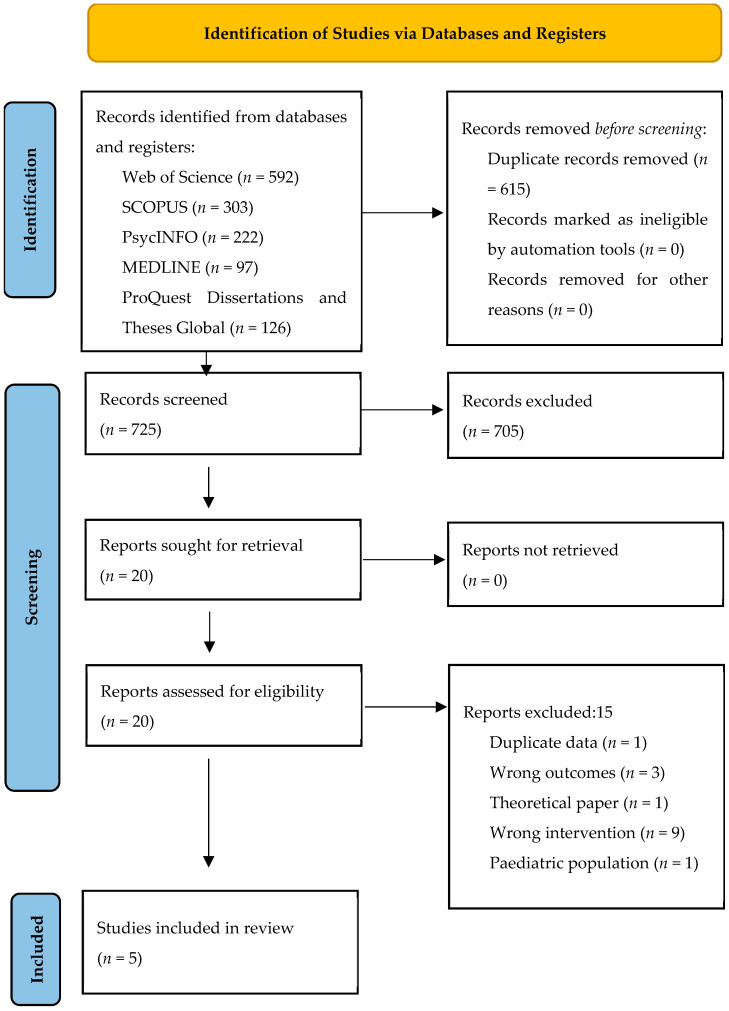
PRISMA flow diagram.

**Table 1 behavsci-15-00119-t001:** List of databases searched and their search fields.

Type	Database	Field(s)	Exclusion
Bibliographic	PsycInfo (Ovid interface)	Title and abstract	Using “Advanced Search”
Bibliographic	MEDLINE (PubMed)	Title, abstract, and keywords	Using “Advanced Search”
Bibliographic	SCOPUS	Title, abstract, and keywords	Using “Basic Search”
Bibliographic	Web of Science	“Topic”	Using “Basic Search”
Theses	ProQuest Dissertations & Theses Global	“NOFT”	Using “Advanced Search”

**Table 2 behavsci-15-00119-t002:** Inclusion and exclusion criteria.

Criteria	Inclusion	Exclusion
Language	Articles published in English	Articles published in other languages
Type of Text	Empirical articles published in peer-reviewed journals or as dissertations Articles must be available as full-text	Preprints, conference abstracts, commentary articles, conceptual reviews, other non-peer-reviewed work
Relevant Population	Studies which examine the effects of meditation interventions on buffering attachment insecurity or enhancing attachment security in adults All geographic locations	Studies conducted on people under the age of 18
Relevant Interventions	Interventions which include either mindfulness or loving-kindness elements as their main premise (including if labelled differently, such as compassion-based interventions) Interventions which include both mindfulness and loving-kindness will be included, but analyzed separately Interventions with any number of arms Interventions must include two or more sessions	Meditation interventions which consist mainly of elements other than mindfulness and/or loving-kindnessMeditation inductions, which consist of only a single session
Relevant Outcomes	Studies which include personal, relational, and health outcomes Studies in which meditation interventions are examined as moderators of attachment orientations in predicting outcomes such as health (e.g., depression, anxiety) or relationship outcomes (e.g., relationship quality, sexual satisfaction) Studies which examine the role of meditation interventions in changing attachment orientations themselves	Studies which focus on state rather than dispositional attachment

**Table 3 behavsci-15-00119-t003:** Summary of study characteristics extracted and coded.

Variables	Details
Citation	APA citation of study
Study Aim	Goal of study
Study Design	e.g., daily diary
Type of intervention(s) used (as categorised in study)	Name(s) of intervention(s) used in study
Type of intervention(s) used (as categorised by the current authors)	Broad type of intervention (e.g., mindfulness, loving-kindness)
Frequency of intervention	Number of sessions
Duration of intervention sessions	e.g., 60 min
Total duration of intervention	e.g., 8 weeks
Duration of study	Total duration of study (including follow-ups)
Number of follow-ups after intervention	e.g., 1
Treatment arms	Number and names of treatment arms, including intervention and control conditions
Type(s) of attachment assessed	e.g., anxiety, avoidance
Outcome(s) assessed	e.g., emotions, relationship satisfaction
Medium of intervention	Online, in-person, audio recordings, or other
Total sample size	Number of study participants
Sample characteristics	Age information, gender distribution, ethnicity distribution, sexual orientation distribution
Anxiety buffered or changed?	Yes/No with potential notes for different outcomes
Avoidance buffered or changed?	Yes/No with potential notes for different outcomes
“Insecure” attachment buffered or changed?	Yes/No with potential notes for different outcomes
Secure attachment changed?	Yes/No
Publication Status	Peer-reviewed journal or dissertation
Geographical region where research was conducted	Africa, Asia, Europe, North America, South America, Antarctica, Australia
Further Notes	Further comments if relevant

**Table 4 behavsci-15-00119-t004:** Extracted data from eligible texts.

Variables	Text 1	Text 2	Text 3	Text 4	Text 5
Citation	[11] ([11])	[16] ([16])	[32] ([32])	[43] ([43])	[54] ([54])
Study Aim	To investigate whether adult attachment orientations moderate the effect of MBSR participation on levels of perceived stress	To test whether a brief mindfulness intervention improved the cognitive, affective, and behavioural aspects of sexual experiences and whether effects varied by attachment orientations	To pilot test a novel trauma-informed model of MBSR	To examine whether different types of mindfulness (internally focused vs. externally focused) were associated with different experimental outcomes	To examine how dimensions of attachment insecurity moderate shifts in positive and negative emotions in response to mindfulness and loving-kindness meditation practices
Study Design	Longitudinal, within-participant	Longitudinal, within-participant, daily diary	Longitudinal, between-group	Longitudinal, between-group, daily diary	Longitudinal, between-group
Type of intervention(s) used (as categorised in study)	MBSR	Mindfulness	Trauma-informed MBSR	Internally focused meditation, externally focused meditation	Mindfulness, loving-kindness
Type of intervention(s) used (as categorised by us)	Mindfulness with elements of loving-kindness	Mindfulness	Mindfulness with elements of loving-kindness	Mindfulness (two types, internally and externally focused)	Mindfulness, loving-kindness
Frequency of intervention	Once a week	Daily	Once a week	Daily	Once a week
Duration of intervention sessions	2.5 h	10 min	2–2.5 h	8 min	1 h
Total duration of intervention	8 weeks	4 weeks	8 weeks	2 weeks	6 weeks
Duration of study	8 weeks	6 weeks	8 weeks	2 weeks	12 weeks
Number of follow-ups after intervention	1	7 (daily)	1	1	Daily for 3 weeks
Treatment arms	1 (MBSR)	1 (Mindfulness intervention)	2 (TI-MBSR and Waitlist Control)	3 (Internally focused meditation, externally focused meditation, control audio condition)	2 (Mindfulness and loving-kindness meditation)
Type(s) of attachment assessed	Anxiety and avoidance measured; secure and insecure categories created	Anxiety and avoidance	Anxiety and avoidance	Anxiety	Anxiety and avoidance
Outcome(s) assessed	Perceived stress	Positive sexual experience, sexual motives, sexual communal strength, unmitigated sexual communal strength	Follow-up attachment and avoidance	Trait mindfulness, relationship satisfaction	Positive and negative emotions
Medium of intervention	In-person	Audio recordings	In-person	Audio recordings	In-person
Total sample size	131	90	45	160	113
Age Information	Secure group: M (SD) = 49.57 (12.99) Insecure group: M (SD) = 47.07 (12.24)	M (SD) = 33.23 (7.71)	M (SD) = 41.5 (14.6)	M (SD) = 20.9 (3.8)	M (SD) = 47.10 (10.49)
Gender Distribution	Secure group: Female = 79%, Male = 20%, Undisclosed = 1% Insecure group: Female = 78%, Male = 20%, Undisclosed = 1%	Female = 93.33%	Female = 100%	Female = 87.5%, Male = 11.9%, Prefer not to disclose = 0.6%	Female = 83.93%
Ethnicity Distribution	Secure group: Asian/Pacific Islander = 1%, White = 97%, Hispanic/Latino = 1%, Undisclosed = 1% Insecure group: African American = 3%, White = 94%, Hispanic/Latino = 2%, Undisclosed = 1%	New Zealand European = 72.22%	TI-MBSR: White European Descent = 79.2%, African Descent = 8.3%, Indigenous to North/South America = 4.2%, Other = 8.3% Waitlist Control: White European Descent = 66.7%, African Descent = 9.5%, Indigenous to North/South America = 9.5%, Hispanic = 4.8%, Other = 4.8%, Would rather not disclose = 4.8%	Caucasian = 70.4%, East Asian = 15%, Multiple Ethnicities = 8.1%, South Asian = 3.1%, Latinx = 1.3%, Southeast Asian = 0.6%, Prefer not to respond = 0.6%	White = 55%, Black = 34%, More than one race = 5%, Asian = 5%, Native Hawaiian or Pacific Islander >1%
Sexual Orientation Distribution	N/A	Heterosexual = 78.89%	N/A	Straight or heterosexual = 65.6%, Equal attraction across genders = 6.9%, exclusively attracted to individuals of the same gender = 0.6% Remaining sample was evenly distributed across different attractions to both genders, identified as asexual, reported a different sexual orientation, or preferred not to respond	N/A
Anxiety buffered or changed?	N/A	Yes, buffered (on positive sexuality)	Yes	Yes, by internally focused meditation on trait mindfulness	Yes, individuals with greater attachment anxiety and randomised to mindfulness meditation reported significant increases over time in positive emotions alongside decreases in negative emotions
Avoidance buffered or changed?	N/A	No	No	N/A	Yes, individuals high in attachment avoidance reported significant decreases in negative emotions in both meditation groups
“Insecure” attachment buffered or changed?	Marginal buffering effect (reported as significant)	N/A	N/A	N/A	N/A
Secure attachment changed?	N/A	N/A	N/A	N/A	N/A
Publication Status	Peer-reviewed journal	Peer-reviewed journal	Peer-reviewed journal	Dissertation	Peer-reviewed journal
Geographical region where research was conducted	North America	Australia	North America	North America	North America
Further Notes	N/A	N/A	N/A	Data were extracted from Experiment 2	N/A

## Data Availability

Our preregistered protocol is available at https://osf.io/qkd65/?view_only=f454bb0f70c847d19523b8705c019e17, accessed on 15 April 2024.

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
