# Peer review of "Mindfulness, Loving-Kindness, and Compassion-Based Meditation Interventions and Adult Attachment Orientations: A Systematic Map"

_behavsci, 2025, doi:10.3390/bs15020119_

Round 1
Reviewer 1 Report
Comments and Suggestions for Authors
Thanks for the Editor for giving me this opportunity to review this article. This is an interesting and potentially important study. This manuscript applied the systematic map, a relatively new method, to review the literatures of mindfulness/meditation interventions on adult attachment.
The majority of my comments are directed at ways the manuscript can be improved. The following points need to be further improved:
(1) In the Introduction section, it would be better to provide a more detailed introduction to the theory and research on the relationship between attachment and mindfulness.
(2) There is a calculation error in Figure 1. In the final step of screening, the total number of excluded literature is not equal to the sum of the specific excluded literatures.
(3) What is “pure” mindfulness intevention? (P17 line294, 299). Nowadays, there are various mindfulness/meditation-based inteventions, but which one can be called “pure” type? Although the authors have provided some explanations and cited literatures, this issue is still quite ambiguous.
(4) In the Discussion section, many of the contents mentioned in Table 1 (e. g. frequency of intervention, duration of intervention sessions, sample size, gender distribution, etc.) have not been specifically discussed.
Reviewer 2 Report
Comments and Suggestions for Authors
1. Summary
The study presents a systematic mapping of the effects of mindfulness, loving-kindness and compassion-based meditation interventions on adult attachment orientations. Five studies were analysed, which suggest that these interventions have a positive impact, especially in people with high levels of attachment anxiety. However, a lack of longitudinal research and cultural diversity in the samples is identified.
2. Overall Assessment
In my opinion, the study is very interesting and aims to provide results that offers a valuable contribution by identifying key gaps in the literature on meditation and attachment interventions. However, the limited number of studies and lack of comparable methodologies limit the strength of the conclusions.
3. Strengths
a) Methodological Rigour: The use of a pre-registered protocol and standardised extraction framework ensures transparency and reproducibility.
b) Identification of Gaps: The study highlights critical areas for future research, such as the need for more longitudinal studies and the inclusion of compassion-based meditations.
c) Theoretical Relevance: The analysis is based on robust theoretical models, such as the Attachment Security Enhancement Model (ASEM).
4. Aspects to Improve:
a) Sample Size and Diversity: Most of the studies were conducted in Western contexts and with a predominance of women, limiting the generalisability of the results.
b) Heterogeneity in Designs: Interventions vary significantly in structure and duration, making comparisons between studies difficult.
c) Lack of Long-Term Follow-Up: Most studies lack long-term measurements, which is crucial for assessing the sustained effects of interventions.
d) Weak Evaluation of Compassion-Based Meditation: There is a lack of studies focusing on this type of intervention.
In conclusion, the paper is a valuable starting point for future research, but requires expansion in both cultural diversity and methodological rigour in future studies.
Round 2
Reviewer 1 Report
Comments and Suggestions for Authors
I am satisfied with the author's response to my comments. All my concerns are addressed in the manuscript R1.